# Porous Alumina Ceramics Obtained by Particles Self-Assembly Combing Freeze Drying Method

**DOI:** 10.3390/ma12060897

**Published:** 2019-03-18

**Authors:** Shujuan Hu, Bo Feng, Xiaoxia Tang, Yue Zhang

**Affiliations:** 1Key Laboratory of Aerospace Materials and Performance (Ministry of Education), School of Materials Science and Engineering, Beihang University, Beijing 100191, China; humenghan2011@126.com (S.H.); bofeng2019@163.com (B.F.); tangxx@buaa.edu.cn (X.T.); 2School of mechanical and chemical engineering, Wuzhou University, Wuzhou 543002, China

**Keywords:** porosity, alumina ceramics, self-assembly, freeze drying

## Abstract

An innovative approach for fabricating porous alumina ceramics is demonstrated in this paper. The distinguished feature is that the construction of the porous structure stems from the interaction between ceramic particles, which is a poorly explored area. By tailoring the Derjaguin-Landau-Verwey-Overbeek (DLVO) interaction energy to the second minimum, the dilute ceramic slurry would be gelled by the weakly assembled particle network, and the assembled structure is conserved via a freeze drying strategy. The DLVO theoretical analyses revealed that the second minimum of interaction energy could be obtained when the counter-ion concentration in colloidal suspension is 1.5 × 10^−2^ mol/L. The properties of the as-assembled samples were compared with one produced by the conventional freeze drying method. Results showed that the self-assembly of alumina particles has a positive influence on micro structures. Unlike the laminar pores generated by the traditional freeze drying procedure, the assembled samples show homogeneously interconnected and hierarchical open pores which were stable even after a 24 h dwell time at 950 °C (open porosity is 79.19% for the slurry of vol 20% solid loading). Particularly, after sintering at 1550 °C for 2 h, open porosity (67.01%) of the assembled samples was significantly greater than that of their un-assembled counterparts (39.97%). Besides, the assembled sample shows a narrower pore size distribution and a relatively higher cumulative pore volume.

## 1. Introduction

Porous ceramics have attracted considerable attention for their excellent thermal shock resistance, low density, low thermal conductivity, high refractoriness, high permeability, high specific surface area, constant filter quality and longer lifetime [1,2,3],thus they are extensively employed in industrial applications including gas filters [4], fluid filtration, sensors, bioreactors, catalytic substrates, bone substitute, insulating lining, random materials, light weight components [5] and so on. All those outstanding performances are closely related to the size, distribution and morphology of the pores, as well as their interconnectivity. Consequently, controlled porous structures, narrow pore size distribution plus a relatively high surface area are of vital importance to the development of porous ceramics [6].

Until now, various ways have been employed for preparing porous ceramics, such as the replica technique, free drying, direct foaming, foam gel-casting, reaction bonding, sacrificial template, gel-casting, pore forming agent (PFA), extrusion, sol-gel method and so on, among which freeze drying [7,8,9] and gel casting [10,11] are preferred due to their simplicity and affordability. Gel-casting is a well-established near-net-shape technology for producing high-quality and complex shaped ceramic parts. The typical route of gel-casting process is dispersing the ceramic powders into an aqueous solution containing the monomer, cross-linker, free radical initiator and catalyst to form the stable and flowing slurry and then pouring the slurry into an appropriately designed mold. By in situ polymerization of the organic monomers added in the slurry, a 3D polymer-water gel network would be formed, within which the dispersed ceramic particles were immobilized. The green bodies prepared via the gel-casting procedure show a higher porosity, homogeneity and reliability, and further, the polymer-water gel is strong enough for machining into precise parts without any distortion. Nevertheless, the main component of the gelling system, e.g., acrylamide (AM), is a neurotoxin [11]. Moreover, the inner stress, micro-cracks and surface-exfoliation phenomenon would be triggered in the green body during the gel-casting process, which will limit its further application and the reliability of ceramic parts to some extent [12].

Another frequently employed technique for preparing porous ceramic materials is free drying [13] which is deemed as an environmentally friendly and pore structure controllable method. The advantages of freeze drying [14] are that the morphology and alignment of the porous structures can be tailored as intended. Via this fabrication technique, porous structures were left by sublimating the solvent, and the micro structure is almost a direct replica of the frozen ice crystals when water is chosen as the dispersive medium [15]. During freezing, the dispersed particles are pushed away, devoured or oppressed by the moving solidification front. Consequently, pore size, final porosity and morphology of the as-obtained porous ceramics can be tuned by freezing kinetics [16]. Whereas porous ceramics obtained from the water based freeze drying method exhibited big and laminar pores due to the growth of ice crystals [17], it was hard to keep the ice crystals in small scales during the solidification process. Researchers have proven that increasing the solid content of ceramic colloidal [18] and/or adding organic polymers [19] could effectively curb the growth of ice crystals. However, the large amounts of polymer adopted in these experiments (ranged from 2 wt % to 10 wt % or even higher) will change the properties of ceramic slurries and complicate the preparation procedures. In addition, the final porosity is closely related to the solid loading of the ceramic slurry, higher porosity can’t be expected from thick slurries. Thus, it is of significant importance to find a route to control pore structure without the addition of polymers when dilute slurry was employed. 

In this work, inspired by the gel-casting method, a novel forming strategy which combined ceramic particle self-assembly and freeze drying was described. Instead of adding organic polymers, the gelation of the slurry was accomplished by ceramic particles self-assembly. Through tailoring the DLVO interaction energy between ceramic particles to the second minimum, the highly dispersed state of ceramic particles in the low concentrated slurry will change into a weakly agglomerative state, and after self-assembly of the loosely agglomerated particles, the deionized water which acts as the dispersive medium will be gelled therein, just like being partitioned into many “rooms” in a “multi-storey building”. Depending on the freeze drying technique, the assembly structure will be conserved and a well interconnected and hierarchically arranged open pore texture (medium sized open cells with a size of several μm and several tens to hundreds of nm pores in the shell of cells) was thus built after a 950 °C heat treatment without introducing any organic polymers. Moreover, the particle-water gel after being assembled can develop sufficient strength to curb the advancement of the moving solidification front and further ensuring the preservation of the original assembled network. Particular emphasis was placed on exploring the experiment condition for attaining the ceramic particles self-assemble by theoretical calculation and comparing the micro structure characteristics (e.g., pore morphology, pore volume, pore size distribution, retention rate of porosity, specific surface area) between the assembled and un-assembled samples based on the experimental results. The incorporated fabrication means proposed in this paper could be identified as an eco-friendly, convenient and cost-effective method to prepare high-performance porous materials from the point of mutual interactions of the ceramic particles, and the resulting hierarchical networked structure is deemed to be a promising candidate for employment in the areas of filtration or catalysis [20]. In general, our work may provide a new and facile method for the preparation of porous ceramics with high porosity, and the prepared ceramics show excellent stability at high temperature.

## 2. Theoretical Background and Calculation 

### 2.1. The Classical Derjaguin-Landau-Verwey-Overbeek (DLVO) Theory

The classical DLVO theory, which was developed in the 1940s by Derjaguin, Landau [21], Verwey and Overbeek [22], is commonly used to describe the interactive forces between particles in an aqueous solution thus elucidating the dispersive or agglomerative state of the particles [23]. In the DLVO theory, the two decisive interactions responsible for the stability of a colloidal system are attractive Van der Waals interactions (*E_A_*) and the repulsive electrostatic interactions (*E_R_*). The strength of the Van der Waals interactions (*E_A_*) is determined by the size of the colloidal particles and by the chemical composition of the system, which is described by the Hamaker constant *A* [24]. The attractive Van der Waals force which is free from the influence of salt concentration, is strong at small separations and gives the dominant contribution to the total energy. The repulsive electrostatic interactions (*E_R_*) which are responsible for the stability of the particles is of an electrostatic nature. With the introduction of inert salts (salts that are composed of ions that do not adsorb on the particles surface), an electrostatic double layer is formed around the charged colloidal particles.

As the schematic drawing (Figure 1) of *E_A_*, *E_R_* and *E_T_* according to the separation distance *h* between particles depicted, with the increase in interval between two particles, the decay velocity of Van der Waals interactions (*E_A_*) is smaller than that of the repulsive electrostatic interactions (*E_R_*). When the gap between themselves is larger enough (*h* > 0′), neither *E_A_* nor *E_R_* is existed and the total energy *E_T_* is zero. Driven by the Brownian motion, particles will close to each other and *E_A_* will come into being firstly and thus resulted a negative *E_T_*. When the interval between particles is getting smaller, the contribution of *E_A_* to *E_T_* is overwhelmed by *E_R_*, the *E_T_* becomes positive gradually and a minimum is thus formed, namely, the second minimum (*a*). However, when the two particles are approaching to certain distance, the influence of *E_R_* is diminished and its dominating position is replaced by *E_A_*. Hence, after building a peak (*E_Max_)* which is called energy barrier, the potential energy curve (*E_T_*) begins to decline and changes into negative eventually. As the distance between two particles is extremely small, a sharp transition occurs to the interaction energy, the Born repulsion resulted from the interaction of the electron cloud, which gives rise to a dramatic rise in *E_T_*, and another minimum *b* called the first minimum is generated consequently.

Whether the colloidal system is stable or not is determined by the size of energy barrier, and particle coagulation will not take place until its energy surpasses *E_Max_*, and thus behaving in coalescing instability. Otherwise, if the energy barrier is so large that the random movement of particles can’t overcome it, the colloidal will present a relatively stable state. As can be seen in Figure 1, the absolute value of the first minimum *b* is far greater than the second minimum *a*, when the particles trapped within the first minimum, the resulted aggregate is dense, stable and irreversible. Whereas, when particles fall into the second minimum, the formed structure is loose, unstable and reversible [25].

### 2.2. Influence of Salt Concentration on DLVO Interaction Energy

One of the most remarkable features of the classical DLVO theory is the presence of a secondary minimum (*a*) in the curve of interaction energy *E_T_* versus particle separation distance *h*. If this minimum is moderately deep in relation to *kT*, it could give rise to the formation of loose flocculation which could be reversed easily. Von Burzagh, Van den Tempel and Kitchener [26] provided experimental evidence in favor of the existence of such a secondary minimum. 

The ability to keep a dispersive or agglomerated state of the charged colloids in water is generally determined by the interplay between attractive Van der Waals (*E_A_*) and repulsive electrostatic interactions (*E_R_*) [27]. This balance can be changed easily from long-ranged repulsion to irreversible aggregation between colloids via tailoring the experimental conditions such as adding salts. By adjusting the overall interactions between colloids, a different coagulative structure could be maintained. Depending on the calculated interaction energies at varied conditions, Zhou et al. reported that the results obtained from the classical DLVO theory were consistent with the experimental dispersion and coagulation behavior of muscovite [28]. Nosrati et al. reported that the interaction energies calculated using DLVO theory were applicable to the predictions of the dispersive or aggregated behavior of colloidal particles [29].

According to classical DLVO theory, the total interaction energy *E_T_* between colloidal particles can be expressed as the sum of the Van der Waals energy (*E_A_*) and the electrostatic energy (*E_R_*), as described by Equation (1) [30].
(1)$$E_{T}{=E}_{A}+E_{R}$$

(1)Van der Waals Interaction

For simplicity, the alumina particles were regarded as two ideal spheres approximately and the measured zeta potential value as the surface potential in this work, and the Van der Waals interaction energy can be calculated using Equation (2).
(2)$$E_{A}=-\frac{A_{121}}{6h}\frac{R_{1}R_{2}}{R_{1}+R_{2}}$$
Where *R*_1_ and *R*_2_ (70 nm) is the radii of two spherical alumina particles respectively. *h* is the separation distance between them. *A*_121_ is the Hamaker constant of alumina particles 1 interacting in medium 2, it can be calculated by Equation (3) [31].
(3)$$A_{121}={\left(\sqrt{A_{11}}-\sqrt{A_{22}}\right)}^{2}$$
where *A*_11_ is the Hamaker constants of the alumina particles and *A*_22_ is the Hamaker constant of water in a vacuum. The Hamaker constants of water and alumina were taken to be 3.68 × 10^−20^ J, 3.3 × 10^−20^ J, respectively [32].

(2)Electrostatic Interaction

The interaction between double layers of spherical particles can be calculated using Equation (4) [31].
(4)$$E_{R}=\frac{\varepsilon_{0}\varepsilon R_{1}R_{2}}{4\left(R_{1}+R_{2}\right)}\left(\varphi_{1}^{2}+\varphi_{2}^{2}\right)\times \left[\frac{2\varphi_{1}\varphi_{2}}{\varphi_{1}^{2}+\varphi_{2}^{2}}Ln\frac{1+\exp\left(-\kappa h\right)}{1-\exp\left(-\kappa h\right)}+Ln\left(1-\exp\left(-2\kappa h\right)\right)\right]\;$$
where $\varphi_{1}$ and $\varphi_{2}$ represent the Stern potentials of the alumina particles which are commonly replaced with the measured zeta potential (65 mV) in this experiment, *ε* is the relative dielectric constant of the dispersive medium (81 for DI water), *ε*_0_ is the dielectric constant of a vacuum (8.854 × 10^−12^ C^2^·m·J^−1^), and 1/*κ* is the Debye length, which can be calculated using Equation (5) [33].
(5)$$\kappa^{-1}=\sqrt{\frac{\varepsilon_{0}\varepsilon k_{B}T}{2N_{A}e^{2}I}}$$
where *N_A_* is Avogadro’s number (6.02 × 10^23^ mol^−1^), $k_{B}$ is Boltzmann constant (1.38 × 10^−23^ J·K^−1^), *T* is Kelvin temperature (298.15 K), *e* is elementary charge (1.602 × 10^−19^ C) and *I* is the ionic strength of the slurry which termed by Equation (6).
(6)$$I=\frac{1}{2}\overset{n}{\underset{i=1}{\sum }}Z_{i}^{2}C_{i}$$
Where $Z_{i}$ and $C_{i}$ are the valence and concentration of the ions, respectively.

## 3. Experimental 

### 3.1. Materials 

To explore the effect of ceramic particles self-assembly on the microstructure of the as obtained porous samples, the commercial α-Al_2_O_3_ (TM-DAR, Taimei Chemicals Co., Tokyo, Japan) with median diameter (D50) of 140 nm and specific surface area of 13.4 m^2^/g was employed as starting material. The strong cationic dispersant poly (dimethyl diallyl ammonium chloride) solution (PDADMAC 35 wt %) was purchased from Sigma Aldrich (Shanghai, china) and used as received. Sodium chloride (NaCl) used as electrolyte to trigger the self-assembly of Al_2_O_3_ particles.

### 3.2. Fabrication Procedure

The Al_2_O_3_ slurry with initial solid loading of 20 vol % were prepared by mixing Al_2_O_3_ powder and dispersant in the distilled water. The slurry was milled with zirconia balls (Zibo yingchi ceramic new material co. LTD, Zibo, China) in the planetary ball mill for 4 h at a rotate speed of 350 r/min. Half of the as-prepared highly dispersed ceramic slurry was further mixed with NaCl (1.5 × 10^−2^ mol/L) for 0.5 h by magnetic stirrer. After assembling for 0.5 h, the assemble slurry and the remnant in former step, which termed as A and N respectively, were casted into cylindrical steel molds with dimension of φ40 × 10 mm. Finally, the two kinds of samples were immersed into liquid nitrogen for quick freezing. The frozen samples were transferred to the lyophilizer (FD-1A-50, Beijing Boyikang, Ltd, Beijing, China) for 36 h to remove the ice crystals. After drying, all the samples were sintered at 950 °C for 24 h with a heating rate of 1 °C/min and then further heated to 1550 °C for 2 h with a heating rate of 10 °C/min, all the sintering procedures were performed in ambient atmosphere.

### 3.3. Tests and Characterization Methods

The rheological property of Al_2_O_3_ slurries were tested using a rotor rheometer (RS/Plus, Brookfield, WI, USA). The zeta potential of the Al_2_O_3_ particles were measured by zeta potential meter (Nano-ZS 90, Malvern Panalytical Ltd, Malvern, UK). Morphologies of porous Al_2_O_3_ ceramics were observed by scanning electron microscopy (FE-SEM, JSM-7500F, JEOL, Akishima, Japan). The porosities of sintered Al_2_O_3_ ceramics were measured by the Archimedes method. The specific surface area, pore size distribution and pore volume were calculated by BET analysis and BJH method (Autosorb-iQ, Quantachrome Instruments, Philadelphia, PA, USA), respectively.

## 4. Results and Discussion

### 4.1. Experimental Conditions Required for the Formation of the Second Minimum a

The open literature reported that the interaction of colloidal particles was extremely influenced by salt concentration in slurry [34,35,36,37,38,39]. Unlike the attractive Van der Waals forces which hardly depended on salt content, the repulsive electrostatic interactions are strongly influenced by it. In the presence of inert salts (salts which are composed of ions that don’t adsorb on the surface of colloidal particles), an electrostatic double layer is formed around a charged colloidal particle. It was proposed that the electrical double layer force could be tailored by controlling the content of salts. Consequently, the total DLVO interaction energy between colloidal particles was altered and thus resulted in different interactive behaviors between them [30].

Inspired by this proposition, the experimental conditions required for the formation of the second minimum of the total DLVO interaction energy between Al_2_O_3_ ceramic particles were explored in this study by controlling the dosage of NaCl. As Figure 2 depicted, at the original dispersive state where the equilibrium concentration of Cl^−^ from the dissociation of dispersant (PDADMAC) is 4.2 × 10^−4^ mol/L, the total DLVO interaction energy between Al_2_O_3_ ceramic particles was positive at the whole separation distance, meaning that the electrostatic interactive energy is far greater than the Van der Waals interactive energy when the distance between Al_2_O_3_ ceramic particles ranged from 0 nm to 80 nm, and the particles in the slurry took on a rather stable dispersive state. Accompanying the addition of NaCl, the total potential energy curve was shrunken owing to the compression effect of Cl^−^ to electrical double layer. Particularly, when the concentration of Cl^−^ increased to 1.0 × 10^−2^ mol/L, the energy barrier decreased to 75 *κ_B_T* from original value of 325 *κ_B_T* approximately. More importantly, as the concentration of Cl^−^ increased to 1.5 × 10^−^^2^ mol/L, a secondary minimum occurred, within which a loose agglomerate of particles could be reversibly formed. However, a total negative interactive energy was obtained when the content of Cl^−^ increased to 1.0 × 10^−^^1^ mol/L, signifying that the repulsive force attributed to the double layer was totally suppressed by the a counter-ion Cl^−^. Particles prefer to attract each other in this case and an irreversible coagulation phenomenon from this unstable system will occur instantly.

### 4.2. Effect of Particle Self-Assembly on Rheological Property of the Slurry

An important method in exploring the rheological property of the alumina slurry is the rheological test, through measuring the viscosity of the slurry, the micro network constructed by alumina particles could be correlated with the presented macro rheological performance. That’s because in such a system, those mechanical properties reflect the microstructure of particles intrinsically, and the viscosity is the result of interactions between ceramic particles. Hence, in order to clarify the effect of particle self-assembly, the viscosity of the ceramic slurry before and after assembly are compared in Figure 3. As depicted in the picture, all slurries had a non-Newtonian behavior and showed a similar shear-thinning phenomenon with the increase in shear rate. The “shear thinning” occurred during the rheological test of the slurry is due to shear disturbances to the suspension structure. At low shear rates state, in order to trigger the flow of the slurry, the particles must flow around or “bounce” between each other, which creates greater resistance therein and a higher viscosity is thus presented in return. When shear rate is increased, the exerted velocity gradient induces the orientation of the particle network which renders the particles to pass through between each other freely at lower shear rate, so the viscosity reduces.

Another case worth mentioned is that the original viscosity of the assembled slurry is distinctly higher than that of the un-assembled ones. Particularly, the starting viscosity of the assembled slurry is as high as 148 Pa·s, which is nearly 15 times higher than its un-assembled counterparts. The main reason accounting for such a phenomenon is that after adding dispersant into the ceramic slurry, the hydrophobic groups of dispersant molecules are adsorbed on the surface of the ceramic particles, and the hydrophilic groups are directed to the liquid phase. Due to the ionization of the hydrophilic polar group, an electrical double layer is formed around the ceramic particles and electrostatic repulsion is thus generated between them. Driven by the electrostatic repulsive force, ceramic particles are separated from each other, the directional flow resulted by shear stress is easily formed in this manner and a lower viscosity is observed in the non-assemble slurry in Figure 3 consequently. On the contrary, for the assembled slurry, the electric double layer is severely compressed by Cl^−^, the interaction energy between themselves is dominated by Van der Waals interactive energy. To destroy the agglomerates constructed by the ceramic particles and render the directional flow, a higher shear stress is inevitably needed, and the macro representation is the dramatic increase in the viscosity of the slurry.

### 4.3. Effect of Particle Self-Assembly on Micro Structure Characteristics of the Porous Ceramics

In order to investigate the difference of pore characteristics between the samples that have undergone and not undergone the assembly procedure (specimens from the assemble slurry were labeled as A and their un-assembled counterparts were labeled as N), the physical properties and microstructure characteristics of the cross-sectioned surface are illustrated by Table 1 and Figure 4, respectively. As elaborated in Table 1, although all specimens after drying possess a similar porosity, porosity retention rate and volume density, which were attributed to the same solid loading of the slurry and drying method, a much higher specific surface area value of ~11.3 m^2^/g was achieved for samples A, almost twice as much as that of samples N (~6.8 m^2^/g). This experiment result further discloses the fact that the specimens experienced the assemble procedure (A) equipped with a more complex and finer micro structure with reference to specimens N. Additionally, when a heat treatment is done, the porosity in all specimens decrease accordingly. Particularly, for samples N, its porosity reduce to 72.96% after sintering at 950 °C for 24 h and this value continue to fall down to 39.97% under the condition of 1550 °C/2 h. However, for samples A, this decrease trend in porosity is rather mild, the sacrifice in porosity after 950 °C/24 h heat treatment is negligible and porosity retention ratio at 1550 °C/2 h is as higher as 67.01%. Moreover, the specific surface area of sample N decreased to 0.02 m^2^/g after sintering at 1550 °C, signifying that the channel wall was totally densified and the micro pore once existed within it were mainly disappeared. While, for the assembled ones, the specific surface area is still as high as 1.71 m^2^/g, proving that certain amount of nanopores still existed in the cell wall. This phenomenon was further certificated by the change in volume density. As can be seen in Table 1, a sharp increase in volume density for the sample N from 950 °C to 1550 °C could be discerned (from 1.07 to 2.37 g/cm^3^) with respect to samples A (from 0.78 to 1.37 g/cm^3^). In a word, all the results obtained from the experiment imply that the micro structure of samples A show better thermo-stability and can endure a much higher service temperature.

The fracture morphologies of assembled and un-assembled specimens are shown in Figure 4. The assembled samples (A) have a well interconnected and hierarchically arranged open pore structure (Figure 4a,c,e). That is, the micron sized pores constructed by assemble ceramic particle agglomerates and nano scaled pores within assemble aggregate itself, which is detailed by Figure 4c. As can be seen in the picture, the assembled sample is stronger enough to withstand the destruction of freeze drying, that is to say, the original structure build by assemble agglomerates is conserved in general. Consequently, a more homogenous 3D reticulated network could be detected as Figure 4a,c,e illustrated. No unidirectional pores or abnormal large pores were found in assembled samples. Because after the ceramic particles’ self-assembly, the viscosity of the slurry increases, which makes the migration of particles to surface of the ice crystals and the growth of ice crystals much harder. Therefore, the samples have a finer microstructure under such a circumstance. Additionally, the deionized water was surrounded by interlocked ceramic agglomerates of which the network was well organized, hence, the growth of ice crystals was restrained in limited scales and resulting in uniform micro pores in the as fabricated porous Al_2_O_3_ ceramics. 

However, for the un-assembled samples N, that’s another story, a more random nature of the lamella alignment porous structure could be found in Figure 4b,d,f. Specifically, as can be seen in Figure 4b, the lamellar architectures behave in a radial manner, and the main contributing reason for the observed phenomenon is that by direct freezing without assembling, it was easy for ice crystals to grow into large unidirectional size since resistance on the ice growth fronts was limited for dilute slurry. Thus, large channel running through the samples were obtained after ice sublimation [40] (see Figure 4d). Moreover, for the dilute slurry contained within the thermal conductive mold is dipped into the liquid nitrogen, when rapid cooling is conducted, solidification is initiated at the perimeter of the suspension, and the resulting ice front propagates horizontally, in a radial manner [41]. The dense layer appeared in the edge of Figure 4b is owing to the fast solidification velocity at the beginning of freeze and suspended particles are initially engulfed by the ice solidification front. Later on, the latent heat released by this early freezing event slows the solidification velocity and particles are subsequently pushed aside by the advanced front rather than engulfed. As solidification progresses, rejected particles are concentrated within laminar space, where the solidification of interstitial fluid eventually occurs as well. Once solidification is completed, the solidified fluid is removed via sublimation, and the pore morphology in the final porous ceramics is a near replica of the morphology of the solidified fluid, and shows radial lamella alignment network throughout the samples.

Also worth mentioning is that an apparent distinction existed in high temperature stability in microstructures between specimen A and N. As can be seen in Figure 4e, after sintering at 1550 °C for 2 h, interconnected open cell constructed by the assemble particles agglomerates was conserved still, and certain amount of porous can be discerned in the walls of open cells. On the contrary, for samples N which experienced the same heat treatment procedure, the particle-packed walls was totally densified, seen in Figure 4f, the contribution to overall porosity left within the samples is macro pore resulted from the sublimation of lamellar ice crystals. This micro information reflected by SEM photo graph is in good agreement with the BET measuring data provided by Table 1. The observed difference in the microstructure between specimens A and N at 1550 °C is owing to different coordination number around the particles. For specimen A, the assembled agglomerates were rather loose, put another way, the number of neighboring particles was small. The grain growth only took place among limited particles and the densification was thus curbed during high temperature sintering procedure. However, for samples N, due to squeezing of ice front, the particles stacked density in the channel walls was rather heavy, meaning that the particle was tightly surrounded by more adjoining particles, consequently, the sintering among particles was facilitated and accelerated, and a larger grain growth phenomenon was discerned in Figure 4f in return.

In order to further reveal the micro characteristics between samples A and N, the nitrogen adsorption tests were conducted on them and the cumulative pore volume and differential pore volume graphs were detailed in Figure 5 and Figure 6 by BJH method. Judging from Figure 5, one can find out that an apparent distinction existed in pore size distribution between the samples A and N. For samples N after drying, the pore size distribution was rather border, ranging from 0 to 200 nm, and this interval reduced to ~80–100 nm when 950 °C heat treatment was subjected to them. The shrinkage in pore diameter distribution range for specimens N was chiefly attributed to the elimination of the small pores and the collapse of the big pores in the wall of channels during high temperature sintering course, which is facilitated by mass transfer process. However, for their assembled counterparts, the pore size distribution is almost keeps the same (~75–135 nm), whether the 950 °C heat treatment was conducted or not. The relationship between cumulative pore volume vs. pore size distribution was depicted in Figure 6. As can be seen in the picture, in comparison with specimens N, specimens A exhibited a much higher cumulative pore volume. Particularly, after sintering at 950 °C for 24 h, its maximum pore volume approaches to 0.12 mL/g, nearly twice the value of N samples under the same condition. What’s more, the contributions of pores smaller than 75 nm or larger than 135 nm to the total cumulative pore volume were negligible, and the pores which made the greatest contribution to the whole pore volume were centrally distributed within ~90–115 nm after drying, and this scope shifted to ~105–135 nm when 950 °C heat treatment was done, which indicated that the assembled samples have a more homogeneous network and the majority of the pore in the as fabricated porous ceramics lay in the interval of ~105–135 nm. The possible reason for this phenomenon is that after tailoring the interactive energy between ceramic particles to the second minimum, agglomerates could be formed by self-assembled ceramic particles. As for the self-assembled ceramic particles driven by a weak Van der Waals attraction, the stacked structure is relatively loose within which the pores were of the same magnitude of particle itself, namely, ~140 nm in this study. 

Furthermore, for the particles in the assembled aggregates already contacted between each other, the collapse of the pore and sintering shrinkage were thus curbed, which positively reasoned the good porosity retention rate (98.99%) listed in Table 1 after a 24 h dwell time at 950 °C in return. Further information could be elicited from Figure 6 in that, after sintering, cumulative pore volume in assembled porous ceramic even downgraded slightly and the size distribution range of the majority pore that contributed the most to the whole cumulative pore volume shifted toward right, which is mainly due to the diffusion of the sub assembled agglomerates surrounded by the primary assemble aggregates. Under the condition of high temperature sintering, these sub assembled agglomerates once acted as obstacle in the cell will move towards their neighboring particles and adhere therein. Consequently, the remaining porous network is getting broader and the total pore volume is reduced to some extent accordingly.

Whereas, for samples N, their cumulative pore volume curves rise in a gentler manner, from zero to its maximum gradually within the full length of the pore size distribution range. As is well known, the pores left within the porous ceramics from un-assembled slurry were just a replica of ice crystals, and during the freezing procedure, ceramic particles were pushed aside by the advancing ice front and squeezed by the ice volume expansion, which facilitated the particles being piled up in a random and heavy manner. As a result, a broader distribution and tiny sizes of the pores were left in the ceramic walls of the channels. Besides, as illustrated in Figure 6, before the 950 °C heat treatment was done, an abrupt change could be observed in the cumulative pore volume curve when pore ranged from ~60 nm to 90 nm, and it shrunk to ~80 nm to 90 nm as a consequence of 950 °C heat treatment, meaning that the pore within the scope of ~60–90 nm occupied the main part of the all after drying and this dominated role was replaced by pores in the interval of ~80–90 nm after sintering. It is important to note that the comparatively narrower pore size distribution of samples N after 950 °C heat treatment doesn’t signify that a much more uniform structure could be obtained from such a technical process, the shrinkage in the pore interval is merely due to the diminishment and/or vanishment of the tiny pore in the wall of the channels under the condition of 950 °C sintering. Moreover, from the cumulative pore volume curve for specimens N, it could be seen that when pore is larger than 100 nm, the pore volume no longer changing basically, signifying that the quantity of pores larger than 100 nm is rather smaller and the majority of the pore in the wall of channel mainly distributed in the interval of ~0–100 nm. 

By taking the two pictures together for consideration, some more precise information concerning the microstructure could be disclosed, that is, compared with the N samples, the one termed as A has bigger cumulative pore volume and narrower pore size distribution. One more case worth mentioning is that the influence of heat treatment on the pore size distribution and cumulative pore volume of A samples is not as obvious as it is on N samples, indicating that the as-obtained porous ceramics via the particle self-assembly procedure show a better high temperature stable performance.

## 5. Conclusions 

In this work, a novel technique that combined ceramic particles self-assembly and freeze drying for the fabrication of highly porous alumina ceramics has been successfully developed. Depending on the theoretical calculation of DLVO interactive energy between the ceramic particles, a loose reticulated structure could be obtained via particle self-assembly when the counter ionic strength in the slurry was tuned to 1.5 × 10^−2^ mol/L, and the constructed network was further conserved by freeze drying. The as-obtained porous alumina ceramics show 3D interconnected and hierarchically arranged open cell morphology. Unlike the specimens from the traditional freeze drying method, the samples which experienced the assembly procedure show higher specific surface area (11.3 m^2^/g after drying), higher open porosity (80% after drying, solid loading of the slurry is 20% vol), higher cumulative pore volume (0.12 mL/g after 950 °C sintering for 24 h) and narrower pore size distribution range (~105–135 nm after 950 °C/24 h sintering). Moreover, the assembled micro structure exhibits a desirable high temperature stable performance. Particularly, its open porosity and specific surface area are 67.01% (pore retention rate is 83.75%) and 1.71 m^2^/g after a 2 h dwell time at 1550 °C, respectively. Above all, this work paves a promising way to fabricate highly porous ceramics based upon the interactive energy between the ceramic particles, and the as-prepared porous ceramics with excellent properties provide potential applications in filtration, catalysis and thermal insulating materials. 

## Figures and Tables

**Figure 1 materials-12-00897-f001:**
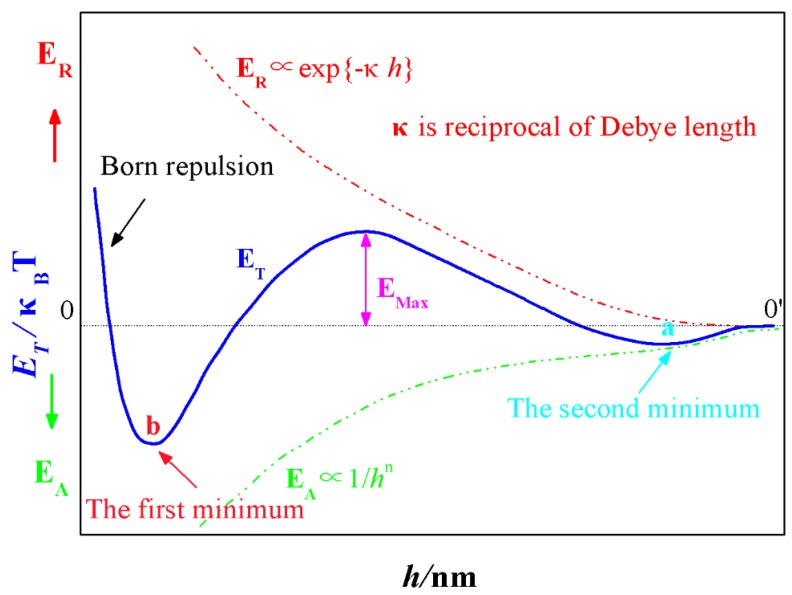
The interaction energy of colloidal particles as a function of their separation distance h.

**Figure 2 materials-12-00897-f002:**
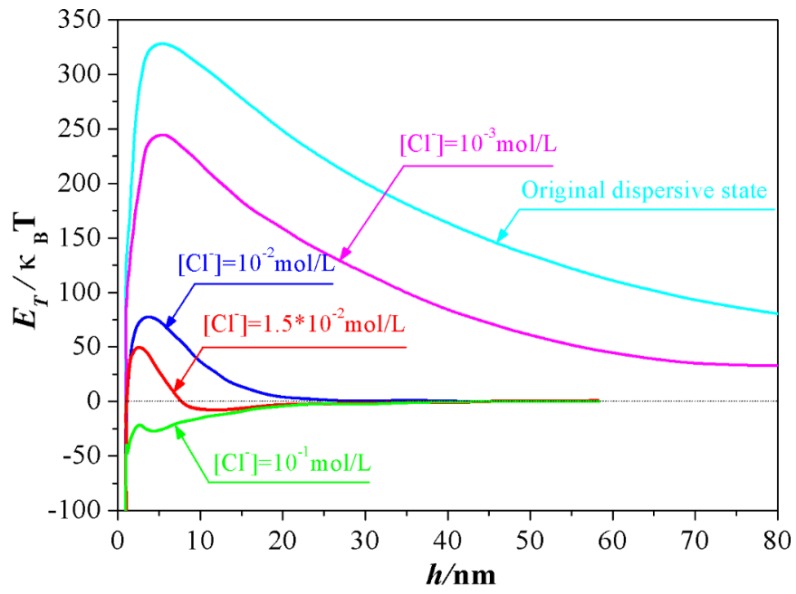
Interaction energy between alumina particles at different ion concentrations.

**Figure 3 materials-12-00897-f003:**
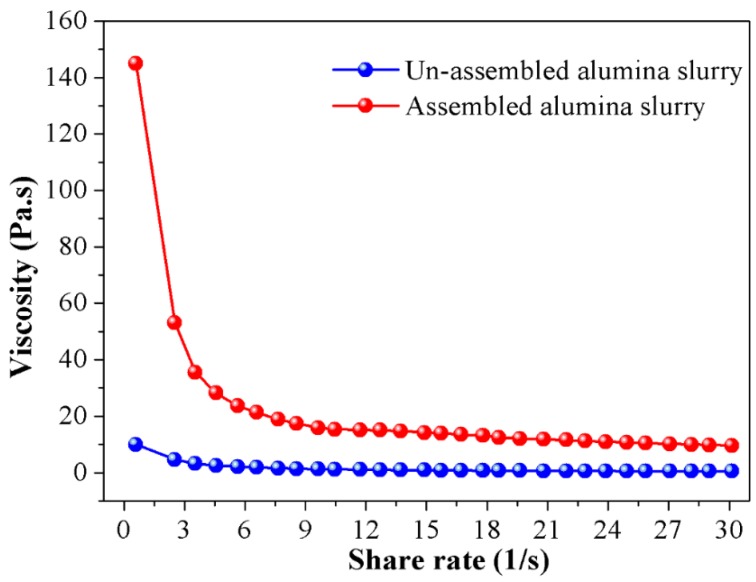
The relationship between viscosity and shear rate for the alumina slurries before and after assembling.

**Figure 4 materials-12-00897-f004:**
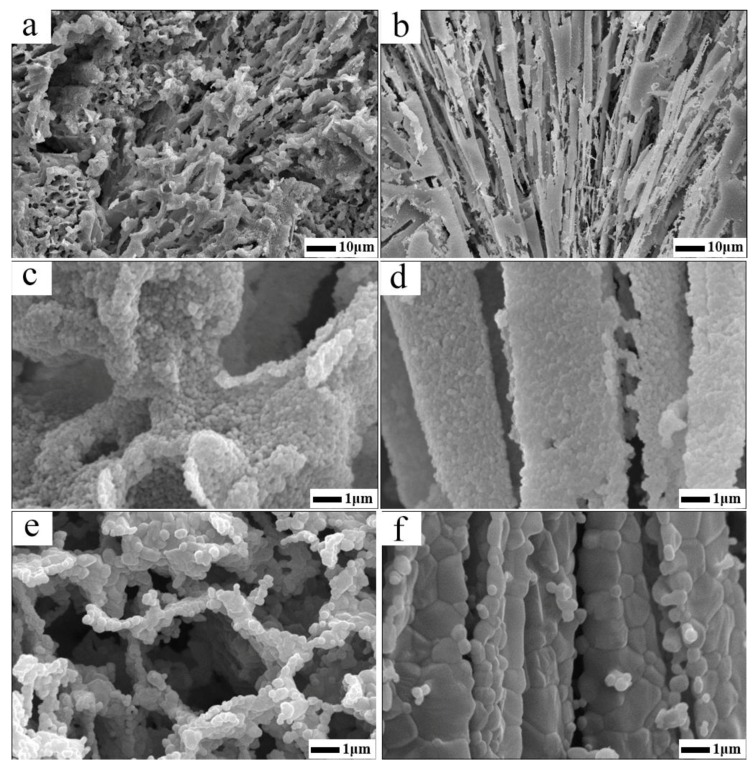
SEM images of the porous alumina ceramics: (**a**) assembled sample after 950 °C heat treatment; (**b**) un-assembled sample after 950 °C heat treatment; (**c**) larger version of a; (**d**) larger version of b; (**e**) assembled sample after 1550 °C heat treatment; (**b**) un-assembled sample after 1550 °C heat treatment.

**Figure 5 materials-12-00897-f005:**
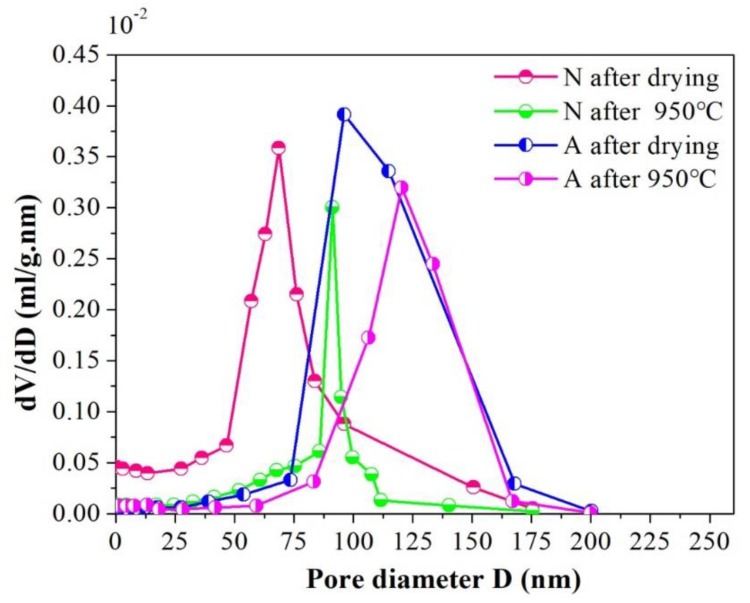
Differential pore volume for samples A and N before and after 950 °C heat treatment.

**Figure 6 materials-12-00897-f006:**
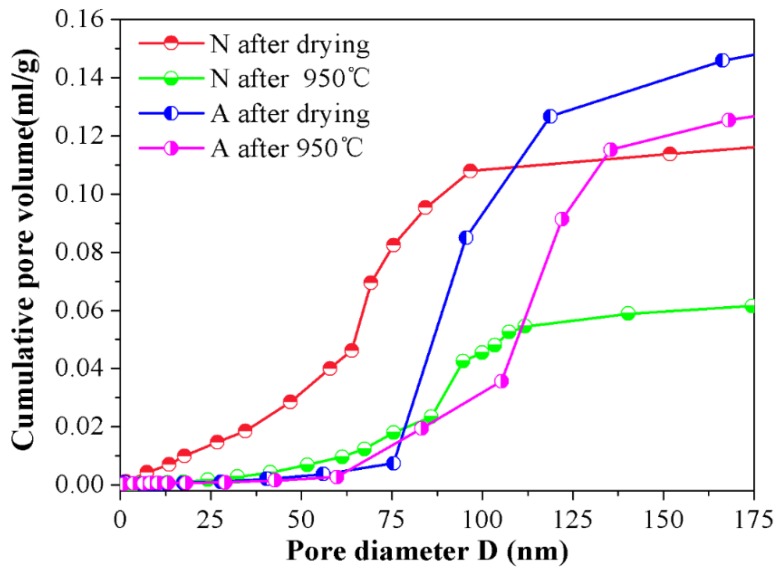
Relationship between cumulative pore volume and pore diameter for samples A and N before and after 950 °C heat treatment.

**Table 1 materials-12-00897-t001:** Physical properties of porous ceramics with and without assembly procedure.

Sample	Pore Characteristics			
	Specific Surface Area (m^2^/g)	Porosity (%)	Porosity Retention Rate (%)	Volume Density (g/cm^3^)
**N-0 °C**	6.8	79.04	98.8	0.83
**N-950 °C**	2.9	72.96	91.2	1.07
**N-1550 °C**	0.02	39.97	49.96	2.37
**A-0 °C**	11.3	80	100	0.72
**A-950 °C**	7.2	79.19	98.99	0.78
**A-1550 °C**	1.71	67.01	83.75	1.37

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
