# Peer review of "Porous Alumina Ceramics Obtained by Particles Self-Assembly Combing Freeze Drying Method"

_materials, 2019, doi:10.3390/ma12060897_

Round 1
Reviewer 1 Report
The manuscript by Hu et al. refers to the study of porous alumina ceramics elaborated by self-assembly and freeze drying techniques. The porous ceramics obtained via this approach are characterized by a 3D interconnected open porosity. The authors showed that the textural properties of the alumina assembled prior to freeze casting are better than the alumina obtained only by tradition freeze casting. They also state that their materials are high temperature stable. In order to optimize the porosity in the alumina, a detailed theoretical calculation of the DLVO interaction energy between alumina particles were conducted. Such materials might pave the way to important energy applications that require good textural properties as well as high temperature stability. However, in view of some imprecisions in the manuscript and errors, the following remarks and comments should be taken into consideration before publishing.
1) First of all, there are a lot of English errors in the MS which makes the reading uncomfortable. Please re-read and correct.
2) The MS is a little bit long; it can be shortened by shortening for example the theoretical analysis in paragraph 2 where there is some repetition.
3) In paragraph 4.1 the font is not homogeneous. Please homogenize.
4) The authors state that the alumina is highly porous; however the results of the specific surface area are relatively low so it is better to say preparation of porous alumina ceramics rather that “highly” porous.
5) Authors are encouraged to show the BET sorption isotherms not only the BJH pore size distribution and pore volume.
6) Authors state that the materials are very stable at high temperature and they explain this by only saying that the porosity is retained at 1550°C, can they please provide further explanation to defend this high temperature stability pf their materials.
Author Response
Detailed Response to Reviewers
We’d like to express our heartfelt thanks to the editor and reviewers for the useful and meaningful comments firstly, which all contribute to making this paper much better.
Below, our responses are in black italic and the corrections to the manuscript are in red italic.
Reviewer #1:
1) First of all, there are a lot of English errors in the MS which makes the reading uncomfortable. Please re-read and correct.
Response: According to your suggestion, we have revised and polished the expression of the whole manuscript carefully, and tried to avoid any illusive sentences and inappropriate presentation. In addition, we have asked some colleagues who are skilled authors of English language papers to better the English for us. We do hope that the language satisfied with the standard of the materials now.
2) The MS is a little bit long; it can be shortened by shortening for example the theoretical analysis in paragraph 2 where there is some repetition.
Response: According to your suggestion, the MS is shortened by deleting several sentences, for example, the sentences in line 108-109 For two similar particles, the Van der Waals forces are always attractive and A is a positive constant; and line 114-118 The simplest description, which holds for monovalent salts at low concentrations, is the Gouy-Chapman solution of the Poisson-Boltzmann equation, which yields that the resulting electrostatic potential decays exponentially with the distance from the colloid. How quick this decay depends on the salt environment and is characterized by the screening length κ-1[24]; and line 136-137 In general, if the EMax is bigger than 15kT, the coagulation caused by collision of irregular thermal motion can be effectively inhibited; and line 210-213 Recently, more and more researchers came to realize that depending on the classical Derjguin-Landau-Verwey-Overbeek (DLVO) theory, not only the colloidal stability could be accounted clearly, but also the interaction behavior between colloidal particles could be elucidated comprehensively[30,34,35,36]are deleted.
3) In paragraph 4.1 the font is not homogeneous. Please homogenize.
Response: Thanks for your careful review and according to your suggestion, the font in paragraph 4.1 has been homogenized already.
4) The authors state that the alumina is highly porous; however the results of the specific surface area are relatively low so it is better to say preparation of porous alumina ceramics rather that “highly” porous.
Response: Thanks for your instructive comment and the adverb highly presented in the whole manuscript is canceled. Besides, the relatively lower specific surface area may be not contradict with high porosity, because the size of the ceramic particle is not too small to some extent, although within the range of nano scale, the size is greater than 100 nm, hence, the specific surface area is not as bigger as that of aerogel.
5) Authors are encouraged to show the BET sorption isotherms not only the BJH pore size distribution and pore volume.
Response: Thanks for your comment. And as for this piece of comment, it would be better for me to give an explanation to you. That is, the obtained BJH pore size distribution and cumulative pore volume are directly derived from the adsorption data, and the information provided by the BJH pore size distribution and pore volume is consistence with that of adsorption isotherm, consequently, I think by providing these two pictures in the manuscript may not influence the completeness of the interpretation. At the same time, the narrower pore size distribution and higher cumulative pore volume of the assembled samples could be highlighted as well. Lastly, if the adsorption isotherm is embraced within the manuscript, not only a repetition would be introduced, but also occupied extra space for publication.
6) Authors state that the materials are very stable at high temperature and they explain this by only saying that the porosity is retained at 1550°C, can they please provide further explanation to defend this high temperature stability of their materials.
Response: Thanks for your comment and the high temperature stability of the porous ceramics fabricated by the method proposed by this research can be reasoned as follows: firstly, as the reviewer pointed out, after sintering at 1550°C for 2 h, the open porosity left within the samples is as high as 67.01% which is much higher than that of the sample non-experienced the assemble procedure (39.97%). Besides, the high temperature stability was further reasoned by its microstructure, which could be found in page 7, line 316-319. Lastly, in order to perfect the explanation in this part, we added the following sentence into the manuscript in the first paragraph of page 7 line 282 to 290, namely, Moreover, the specific surface area of sample N decreased to 0.02 m2/g after sintering at 1550°C, signifying that the channel wall almost completely densified and the micro pore once existed within it were mainly disappeared. While, for the assembled ones, the specific surface area still as high as 1.71 m2/g, certificating that certain amount of nano pores still existed in the cell wall. This phenomenon was further verified by the change in volume density. As can be seen in table 1, a sharp increase in volume density for the sample N from 950°C to 1550°C could be discerned (from 1.07 to 2.37 g/cm3) with respect to samples A (from 0.78 to 1.37 g/cm3). In a word, all the results obtained from the experiment imply that the micro structure of samples A show better thermo-stability and can endure a much higher service temperature.
Reviewer 2 Report
The paper describes the use of the ice-templating technique applied to a loose self-assembled alumina slurry. Firstly, the conditions to obtain the self-assembly of the ceramic particles were studied and then the process was applied to the self-assembled slurry and to a non-assembled slurry. The samples obtained were then characterized and compared.
The topic of the paper is interesting but some points need to be clarified by the Authors, as outlined below:
1. Moderate English changes are required. There are numerous spelling errors along the text.
2. The abstract is too long since it exceed the maximum of 200 words.
3. The references in the text [] must not be reported as apexes.
4. In the text report Figure and not Fig.
5. The keyword “catalysis” is misleading since in the text the catalysis has not been addressed.
6. Page 1/line 12 – the abbreviation DLVO must be defined in parentheses since it is the first time that is introduced.
7. The references are not reported in the format required by the Journal.
8. The “Author contributions” and “Conflict of interest” are missing.
9. In Table 1 the decimal units are not uniform.
10. Page 6/lines 214-220 – the text format is different.
11. Page 5/line 166 – the radius of the particle should be 70 nm if the median diameter reported for Al2O3 is 140 nm.
12. Page 5/line 187 – introduce here the abbreviation PDADMAC and not at line 191.
13. Page 5/line 195 – were the molds fully filled up to 50 mm height? Were the samples totally immersed into liquid nitrogen or just the bottom part?
14. The Authors did a theoretical calculation of DLVO. Did you experimentally try and verify the other suspensions at different ion concentrations?
15. Page 8/line 279 – What do you mean by “porosity retention” and how did you calculate it? If you used the initial porosity and the porosity after the thermal treatment, why the porosity retention of N-0°C is 98.8%?
15. In which part of the samples are taken the sections reported in Figure 4?
16. Which are the final dimensions of the samples? Did you observe different shrinkages in samples A and N?
17. Why the porosity analysis (pore size distribution and pore volume) was not performed on samples treated at 1550°C?
18. Page 2/line 66 – “Unfortunately, porous ceramics obtained from water based freeze casting method exhibited big and laminar pores due to the growth of ice crystals [17] and it was hard to keep the ice crystals in small scales during solidification process.” I think that this is one of the main advantage of freeze casting, not a drawback.
19. Page 8/line 277-281 – this sentence is not clearly expressed.
20. Page 8/line 295 – during the freeze drying there is not a destruction.
21. Page 10-12 – Check well the English.
Author Response
Reviewer #2:
Comments and Suggestions for Authors
The paper describes the use of the ice-templating technique applied to a loose self-assembled alumina slurry. Firstly, the conditions to obtain the self-assembly of the ceramic particles were studied and then the process was applied to the self-assembled slurry and to a non-assembled slurry. The samples obtained were then characterized and compared.
The topic of the paper is interesting but some points need to be clarified by the Authors, as outlined below:
1. Moderate English changes are required. There are numerous spelling errors along the text.
Response: According to your suggestion, we have revised and polished the expression of the whole manuscript carefully, and tried to avoid any illusive sentences and inappropriate presentation. In addition, we have asked some colleagues who are skilled authors of English language papers to better the English for us. We do hope that the language satisfied with the standard of the materials now.
2. The abstract is too long since it exceed the maximum of 200 words.
Response: Thanks for your kind suggestion and directed by your comment, the abstract has been shortened as follows: An innovative approach for fabricating porous alumina ceramics is demonstrated in this paper. The distinguished feature is that the construction of the porous structure stems from the interaction between ceramic particles, which is poorly explored. By tailoring the DLVO interaction energy to the second minimum, the dilute ceramic slurry would be gelled by the weakly assembled particle network and the assembled structure is conserved via freeze drying strategy. The DLVO theoretical analyses revealed that the second minimum of interaction energy could be obtained when the counter-ion concentration in colloidal suspension is 1.5×10-2 mol/L. The properties of the as-assembled samples were compared with the one that produced by conventional freeze drying method. Results showed that self-assembly of alumina particles has a positive influence on micro structures, unlike the laminar pore generated by traditionally freeze drying procedure, the assembled samples show homogeneously interconnected and hierarchical open pores which were even stable after a 24 h dwell time at 950 °C (open porosity is 79.19% for the slurry of 20vol% solid loading). Particularly, after sintering at 1550°C for 2 h, its open porosity (67.01%) is significantly greater than their un-assemble counterparts (39.97%). Besides, the assembled one shows a narrower pore size distribution and higher cumulative pore volume relatively.
3. The references in the text [] must not be reported as apexes.
Response: Thanks for your suggestion and directed by your comment, all the writing form of the reference in the text has been changed as follows: constant filter quality and longer lifetimes [1-3]
4. In the text report Figure and not Fig.
Response: Thanks for your suggestion and directed by your comment, all the Fig in the text has been replaced by Figure.
5. The keyword “catalysis” is misleading since in the text the catalysis has not been addressed.
Response: Thanks for your useful suggestion and directed by your comment, keyword “catalysis” is canceled.
6. Page 1/line 12 – the abbreviation DLVO must be defined in parentheses since it is the first time that is introduced.
Response: Thanks for your useful comments and immediately after DLVO, the expression of (Derjaguin-Landau-Verwey-Overbeek) was supplemented already.
7. The references are not reported in the format required by the Journal.
Response: Thanks for your useful comments and the reference format has been changed according to the requirement of the journal.
8. The “Author contributions” and “Conflict of interest” are missing.
Response: Thanks for your useful comments and the “Author contributions” and “Conflict of interest” file will be supplemented.
9. In Table 1 the decimal units are not uniform.
Response: Thanks for your useful comments. Frankly speaking, the units of volume density do uniform with that of others. However, in order to make the readers easily in weighing the value of it and thus judging whether it is dense or not, I think it would be better for me to express in this manner. So, I beg your pardon and your understanding and support.
10. Page 5/line 166 – the radius of the particle should be 70 nm if the median diameter reported for Al2O3 is 140 nm.
Response: Thanks for your useful comments and I have revised the manuscript according to your suggestion. Namely, Where R1 and R2 (70 nm) is the radii of two spherical alumina particles respectively.
11. Page 5/line 187 – introduce here the abbreviation PDADMAC and not at line 191.
Response: Thanks for your useful comments and I have revised the manuscript according to your suggestion. Namely, The strong cationic dispersant polydimethyl diallyl ammonium chloride (PDADMAC) (35wt%) was purchased from Sigma Aldrich.
12. Page 5/line 195 – were the molds fully filled up to 50 mm height? Were the samples totally immersed into liquid nitrogen or just the bottom part?
Response: Thanks for your useful comments. And I am really sorry for the miswriting occurred in the manuscript, cylindrical steel molds with dimension of φ40×10 mm, not 50mm in height, and I have revised it already. And the molds were fully filled up to 10mm height and the whole sample was immersed into liquid nitrogen for freezing.
13. The Authors did a theoretical calculation of DLVO. Did you experimentally try and verify the other suspensions at different ion concentrations?
Response: Thanks for your useful comments. This piece of suggestion is of great important, the suggestion given by you is just the research focus now and I think I will disclosure the related findings in the near future.
14. Page 8/line 279 – What do you mean by “porosity retention” and how did you calculate it? If you used the initial porosity and the porosity after the thermal treatment, why the porosity retention of N-0°C is 98.8%?
Response: Thanks for your useful comments. The porosity retention is just the remain ratio of the porosity, the calculation equation is as below: . And the reason for the porosity retention of N-0°C is 98.8% is that as the solid loading of 20vol%, the porosity in the sample will be as high as 80% if no shrinkage is available. The value of 98.8% means that there is a drying shrinkage occurred within the sample.
15. In which part of the samples are taken the sections reported in Figure 4?
Response: The section was taken from middle part of the sample, however, the position does not impose an influence on the microstructure. No distinct difference was discerned among the sections taken from the top, middle and bottom of the sample.
16. Which are the final dimensions of the samples? Did you observe different shrinkages in samples A and N?
Response: The shrinkage occurred during drying and 950°C heat treatment process were not so obvious, however, after sintering at 1550°C, the shrinkage phenomenon in N samples was much more obvious.
17. Why the porosity analysis (pore size distribution and pore volume) was not performed on samples treated at 1550°C?
Response: Thanks for your useful comments. For the N sample, after sintering at 1550°C, the pore within nano scale is basically disappeared driven by the densification process and the remained pore within the sample are the macro pore generated by the ice, for such bigger pore, the BET method in characterizing the pore structure was no longer suitable. Hence, the porosity analysis depending on BET method was performed in the condition of at 950°C heatment.
18. Page 2/line 66 – “Unfortunately, porous ceramics obtained from water based freeze casting method exhibited big and laminar pores due to the growth of ice crystals [17] and it was hard to keep the ice crystals in small scales during solidification process.” I think that this is one of the main advantage of freeze casting, not a drawback.
Response: Thanks for your useful comments. As you mentioned, that are really not drawbacks for freeze drying, but the characteristic, and according to your suggestion, we have replaced the unfortunately by whereas.
19. Page 8/line 277-281 – this sentence is not clearly expressed.
Response: Thanks for your useful comments and I have revised the manuscript according to your suggestion. Namely, This experiment result further discloses the fact that the specimens A show a more complex and finer micro structure with reference to specimens N. Additionally, when a heat treatment is conducted, the porosity in all specimens decrease accordingly. Particularly, for the samples N, its porosity reduce to 72.96% after 950°C/24 h sintering and this value continually decrease to 39.97% under the condition of 1550°C/2 h.
20. Page 8/line 295 – during the freeze drying there is not a destruction.
21. Page 10-12 – Check well the English.
Response: According to your suggestion, we have revised and polished the expression of the whole manuscript carefully, and tried to avoid any illusive sentences and inappropriate presentation. In addition, we have asked some colleagues who are skilled authors of English language papers to better the English for us. We do hope that the language satisfied with the standard of the materials now.

Round 2
Reviewer 1 Report
Thank you for the replies provided. I re-erad the MS, it was celarly imprved but please before publishing there is still a need for spell check.
Author Response
Detailed Response to Reviewers
We’d like to express our heartfelt thanks to the editor and reviewers for the useful and meaningful comments firstly, which all contribute to making this paper much better.
Below, our responses are in black italic.
Reviewer #1:
1) Thank you for the replies provided. I read the MS, it was clearly improved but please before publishing there is still a need for spell check.
Response: According to your suggestion, we have checked and polished the expression of the whole manuscript carefully, and tried to avoid any illusive sentences and inappropriate presentation. We do hope that the language satisfied with the standard of the materials now.